# A Review on Secondary Immune Thrombocytopenia in Malaysia

**DOI:** 10.3390/healthcare10010038

**Published:** 2021-12-26

**Authors:** Muhamad Aidil Zahidin, Noor Haslina Mohd Noor, Muhammad Farid Johan, Abu Dzarr Abdullah, Zefarina Zulkafli, Hisham Atan Edinur

**Affiliations:** 1Department of Haematology, School of Medical Sciences, Health Campus, Universiti Sains Malaysia, Kubang Kerian 16150, Kelantan, Malaysia; aiditarsier@gmail.com (M.A.Z.); faridjohan@usm.my (M.F.J.); 2Transfusion Medicine Unit, Hospital Universiti Sains Malaysia, Kubang Kerian 16150, Kelantan, Malaysia; 3Department of Medicine, School of Medical Sciences, Health Campus, Universiti Sains Malaysia, Kubang Kerian 16150, Kelantan, Malaysia; abudzarr@usm.my; 4Forensic Science Programme, School of Health Sciences, Health Campus, Universiti Sains Malaysia, Kubang Kerian 16150, Kelantan, Malaysia; edinur@usm.my

**Keywords:** secondary immune thrombocytopenia, autoimmune disease, malignancy, infection, clinical practice guideline

## Abstract

Immune thrombocytopenia (ITP) is an acquired autoimmune disease that occurs in adults and children. In Malaysia, the clinical practice guideline (CPG) for the management of ITP was issued in 2006, which focused almost exclusively on primary ITP (pITP), and only a few secondary ITP (sITP) forms were addressed. All published (twenty-three) sITP articles among children and adults in Malaysia, identified on the academic databases were retrieved. The articles were published between 1981 and 2019, at a rate of 0.62 publications per year. The publications were considered low and mainly focused on rare presentation and followed-up of secondary diseases. This review revealed that sITP in Malaysia is commonly associated with autoimmune diseases (Evan’s syndrome, SLE and WAS), malignancy (Kaposi’s sarcoma and breast cancer) and infection (dengue haemorrhagic fever, *Helicobacter pylori* and hepatitis C virus). The relationship between ITP and autoimmune diseases, malignancy and infections raise the question concerning the mechanism involved in these associations. Further studies should be conducted to bridge the current knowledge gap, and the further information is required to update the existing CPG of management of ITP in Malaysia.

## 1. Introduction

Immune thrombocytopenia (ITP) is an immune-mediated and acquired illness defined by a temporary or permanent reduction in platelet counts, and depending on the degree of thrombocytopenia, it may result in an increased risk of bleeding [1,2,3]. An individual diagnosed with primary ITP (pITP) has a peripheral blood platelet count of <100 × 10^9^/L in the absence of other thrombocytopenia-related causes. The diagnosis of pITP is still based on exclusion; there are presently no reliable clinical or laboratory criteria for accurate diagnoses [1,2]. ITP can be secondary to medications or concurrent diseases such as an autoimmune condition (e.g., systemic lupus erythematosus (SLE), antiphospholipid antibody syndrome (APS) or Evan’s syndrome), lymphoproliferative disease (e.g., chronic lymphocytic leukaemia or large granular T-lymphocyte lymphocytic leukaemia), or chronic infection (e.g., with *Helicobacter pylori* (*H. pylori*), human immunodeficiency virus (HIV) or hepatitis C virus (HCV)) [4,5].

In 2006, the Ministry of Health (MOH) Malaysia and Academy of Medicine Malaysia issued a clinical practice guideline (CPG) for the management of ITP [1]. The CPG focused solely on pITP and was based on expert view, the American Society of Haematology (ASH) and British Society of Haematology (BSH) guidelines, and the results of a comprehensive assessment of current medical literature. CPG also provided a systematic approach to initial diagnoses and disease management, depending on the circumstances linked with ITP. The health care practitioners are accountable for the identification of secondary forms in the diagnostic criteria of pITP [4].

Numerous case reports have been published to address the possibility of secondary forms as underlying or masking diseases that provide a clinical presentation such as ITP. Those findings can update the current knowledge of secondary ITP (sITP) and provide an understanding of the literature and research available that addresses the epidemiology of sITPs in Malaysia.

## 2. Secondary Immune Thrombocytopenia Reported in Malaysia

In the past four decades, 23 sITP literatures were published between August 1981 and December 2019 (Figure 1). The number of publications was considered low (0.62 publications per year) compared to the number of other autoimmune diseases such as SLE and rheumatoid arthritis (based on our surveys using several search engines; i.e., Web of Science, SCOPUS and ScienceDirect). We understand that our country’s health institutions have a data registry for both pITP and sITP cases. Nevertheless, those data have not yet been published, thus causing the true incidence or prevalence of ITP in Malaysia to remain unknown.

Previous sITP studies are rare for ITP presentation and for a follow-up of secondary diseases associated with ITP such as Kaposi’s sarcoma, dengue haemorrhagic fever (DHV), Evan’s syndrome, Wiskott-Aldrich syndrome (WAS), and infections as shown in Table 1. We identified that at least 15 diseases were associated with ITP. Due to the lack low number of publications, we only selected and discussed the top three groups of conditions, which are autoimmune diseases [6,7,8,9,10,11,12,13,14,15], malignancy [16,17,18] and infection [19,20,21,22,23].

## 3. Immune Thrombocytopenia and Autoimmune Diseases

### 3.1. Evan’s Syndrome

Evan’s syndrome is a relatively uncommon autoimmune disorder in which the immune system attacks its own red blood cell and platelet. Affected people commonly experience thrombocytopenia and autoimmune haemolytic anaemia (AIHA), which develop simultaneously or sequentially [29]. Evan’s syndrome is a chronic disease that affects both adults and children [30,31,32]. We identified two cross-sectional studies [6,7] and two case studies [8,9] of Evan’s syndrome between 1991 and 2012. Ng [6] reported on 12 patients with Evan’s syndrome, while Palaniappan & Ramanaidu [9] presented a rare presentation of autoimmune hepatitis overlapping with AIHA and ITP in a male patient. The remaining two works of literature contrarily discussed ITP in relation to splenectomy [7,8].

Previously, the incidence and prevalence of Evan’s syndrome were unknown. That is, until recently, as Mannering et al. [30] determined the prevalence and prognosis of Evan’s syndrome in Danish children under the age of 13. They reported the prevalence from 0.5 to 1.2 per million people, and from 6.7 to 19.3 per million in 1990 and 2015, respectively. On the other hand, the yearly incidence of Evan’s syndrome and its prevalence in Danish adults increased considerably, reaching 1.8 per million-person in 2016 [31]. The epidemiology of Evan’s syndrome in Malaysia remains unknown. However, for the past four decades, 17 patients were described, of whom 85.71% were female (Table 2).

Ten patients (62.50%) were diagnosed with ITP and direct antiglobulin (DAT) positive AIHA simultaneously, while six were later found to be DAT positive. Hamidah et al. [8] revealed that an AIHA-positive patient was diagnosed with ITP seven months after the initial investigation. There was a possibility that patients were antinuclear antibody (ANA) and collagen antibody positive, but there was no other evidence of SLE [6,7]. In Evan’s syndrome, the peripheral blood platelet count ranged from severe thrombocytopenia (<20 × 10^9^/L) to a normal platelet count (150–450 × 10^9^/L) [6,8]. In some cases, thrombocytopenia can be misdiagnosed as non-haemorrhagic stroke and thrombotic thrombocytopenic purpura (TTP) [33,34].

The therapy for Evan’s syndrome can be challenging. The treatment is based on the symptoms that patients present. Prednisolone is the most common first-line treatment, with 25.0% of patients responding to steroid therapy (Table 3). The patients responded well and had mild thrombocytopenia on low dose prednisolone during 1 to 66 months (mean = 28.33) from the last follow-up [6,9]. Danazol, methylprednisolone and splenectomy were considered second-line therapy. A quarter of the patients achieved a long remission up to 106 months (mean = 84.40). Despite this, some patients developed nephrotic syndrome, benign intracranial hypertension and autoimmune hepatitis, which have rarely been reported in previous study [35,36].

Jackson et al. [7] reported that a patient refused splenectomy, yet underwent prednisolone therapy for 38 months. The patient had no bleeding and responded well to the treatment, but developed mild proteinuria. The concurrence of Evan’s syndrome with proteinuria is relatively infrequent. However, it has been reported previously along with other comorbidities [37,38]. Three Evan’s syndrome patients died of large intracranial haemorrhage (ICH) and one due to pulmonary embolism [6]. ICH occurring in patients with ITP and AIHA can be treated using high dose conventional steroids. One of the patients with ICH was reported to have pneumonia, which is relatively uncommon in Evan’s syndrome. The complication of ICH has also occurred in children, and the patient was successfully treated through unrelated cord blood transplantation [39].

### 3.2. Systemic Lupus Erythematosus

SLE or lupus is a chronic autoimmune inflammatory disease that can affect almost any organ in the human body [40]. SLE is more common in women than in males and children, and it is common among Asian and Afro-Caribbean people than in Caucasians [40,41,42]. A number of previous studies were conducted on SLE in terms of prevalence, population distribution, clinical features, and disease expression since 1948. Previous studies identified thrombocytopenia as one of the comorbid diseases in SLE [43,44,45]. We identified two case studies [10,12] and two cross-sectional studies [7,11] of SLE between 1994 and 2015. Jackson et al. [7] described treatments given to a patient who refused splenectomy, while Mohd Shahrir et al. [11] used CD 20 monoclonal antibody in treating severe SLE patients.

According to Malaysia SLE Association (http://lupusmalaysia.org/ (accessed on 10 September 2021)), it is estimated that 10,000 people were affected with SLE in the past three decades. Previously, it was estimated that the prevalence of SLE in Malaysia was 43 per 100,000 in the population, for which the majority were woman (12:1) and the Chinese population (54.8%) [44,45]. There were five cases of ITP with SLE reported from 1994 to 2015 (Table 4). The patients’ age ranged from 8 to 45 years old (mean = 25.33). Currently, there are no specific tests to diagnose SLE. However, the diagnosis can be made based on clinical manifestations, neuropsychiatric disorders, haematological disorders and the detection of ANA and antibodies for double-stranded DNA (anti-dsDNA) [45]. Clinical manifestations include malar rash, oral ulcers, petechial haemorrhages and bruises. Malar rash and molar ulcer are the most reported symptoms of SLE [43,44,45,46,47,48].

The objective of the treatment is to reduce the severity of symptoms based on the parts of the body affected by SLE. Anti-malarial, local agents and non-steroidal anti-inflammatory drugs commonly treat mild lupus symptoms [40]. In the acute phase, glucocorticoids were used to treat individuals with moderate or severe illness. Corticosteroids were used to minimize and suppress the immune response [7,10,11,12,40,43]. The patients responded well to the therapy, as the platelet count had risen and remained normal for up to 52 months (mean = 11.87) [7,10,11,12].

### 3.3. Wiskott-Aldrich Syndrome (WAS)

WAS is a rare X-linked primary immunodeficiency characterised by eczema, micro thrombocytopenia, progressive T- and B-cell immunodeficiency and recurrent infections. The disease has an increased risk of autoimmunity and malignancies. Primary immunodeficiency keeps the immune system from functioning properly and is associated with megakaryocytes suppression, thrombocytopenia, and bleeding. The syndrome is estimated to occur in 1 per 100,000 live male births and 1 to 10 per million children in Norway [49] and the United States and Europe [50], respectively. However, the incidence of WAS in Malaysia is unknown.

We found a single report of WAS that presented with ITP in a 9-month-old boy [15]. He presented with persistent thrombocytopenia and negative antiplatelet antibody (APA) assay. A WAS gene analysis was conducted and revealed a positive result. WAS, in general, was reported as difficult to differentiate from ITP [50,51,52,53]. It could be due to the loss of functional mutations in the WAS gene located in the X chromosome (position Xp11.22-p11.23), and the pathogenic mutations that occur within the exons and introns [54,55]. As previously reported, at least 49 mutations (either nonsense, missense, frameshift, splice site or complex) occurred in the WAS gene [15,50,51,52,56,57,58,59].

## 4. Immune Thrombocytopenia and Malignancy

### 4.1. Kaposi’s Sarcoma

Kaposi’s sarcoma is a malignancy that develops in the lining of lymph and blood vessels. Kaposi’s’ sarcoma tumours or lesions usually develop as painless purple patches on the face, legs or foot. Lesions in the mouth, vaginal region, lymph nodes and digestive tract (severe) are also possible. The root cause of Kaposi’s sarcoma is an infection of human herpesvirus 8 (HHV-8) and occurs in immune-suppressed patients. Kaposi’s sarcoma is more common in patients with HIV infection. However, the mucocutaneous manifestations of Kaposi’s sarcoma are rare [60].

Jing and Ismail [61] reported a low prevalence of Kaposi’s sarcoma incidence in Malaysia. Kaposi’s sarcoma was previously reported to be associated with ITP [16], HIV/AIDS-related [60,61,62] and non-related disease in the country [63,64,65]. The lesions appeared over the face, scalp, hard palate, trunk, and genital area. Noor Akmal and Abdul Wahab [16] reported findings on a patient who developed Kaposi’s sarcoma during corticosteroid therapy for ITP. Choi et al. reported that Kaposi’s sarcoma occurred during immunosuppressive medication, including corticosteroid therapy for ITP, which has an unpredictably poor prognosis [66].

### 4.2. Breast Cancer

Breast cancer primarily affects women, with 1.8 million incident cases, but was also found to affect 1.0% of men in 2013 [67]. Breast cancer was the largest cause of disability-adjusted life years (DALYs) in women [67]. In 2013, developing countries had higher age-standardised incidence rates (ASIRs) per 100,000 and age-standardised death rates (ASDRs) per 100,000 for both sexes than in developed countries [67].

ITP has been described in patients with breast cancer [68,69,70]. In Malaysia, Abdul Wahid et al. [17] reported cases of breast carcinoma presenting as ITP. The ITP was diagnosed when the patient presented with gingival and conjunctival bleeding, and had a platelet count 2.0 × 10^9^/L after excluding drug-induced thrombocytopenia. The cause of the ITP is probably related to immune-mediated thrombocytopenia in view of large platelet in the peripheral blood file and increase numbers of megakaryocytes in the bone marrow [17]. Solid tumour/malignancies are rarely associated with ITP [17,68,69,70]. The association between ITP and solid tumour/malignancy can be caused by a wide range of mechanisms, including marrow hypoplasia due to chemotherapy, tumour infiltration in the bone marrow and platelet consumption due to coagulopathy [71,72].

## 5. Immune Thrombocytopenia and Infection Diseases

### 5.1. Dengue Haemorrhage Fever

Dengue is one of the most dangerous arthropod-borne viral diseases, with a high morbidity and fatality rate. The virus contains four serotypes (DEN 1, DEN 2, DEN 3 and DEN 4), carried by female Aedes aegypti and transmitted from person to person. The dengue incidence has grown dramatically around the world as well in Malaysia. The country experienced 361 cases per 100,000 adults and an incidence rate of 390.4 cases per 100,000 children in 2014 and 2019, respectively [73,74].

Dengue fever is characterised by an acute high fever 3–14 days after being infected. Infected individuals may experience retro-orbital pain, frontal headache, haemorrhagic manifestations, myalgias, rash, arthralgias and leukopenia [73,74,75]. Dengue haemorrhagic fever (DHV) is characterised by the haemorrhagic manifestation, thrombocytopenia and increased vascular permeability [75]. Dengue haemorrhagic fever is comm only found among adult ITP patients [76,77], but a rare cause for paediatric [78]. In Malaysia, there were two reported cases of ITP following dengue infection in paediatric [19,20]. Both patients had severe thrombocytopenia (<10 × 10^9^/L) and showed ITP manifestations during early diagnosis.

### 5.2. Helicobacter pylori Infection

*H. pylori* is a bacterium that causes stomach ulcers. The bacteria enter human body, remains in the intestines and causes ulcers in the stomach lining or the upper part of the small intestine. In some cases, infection can develop into stomach cancer. Various studies investigated the prevalence of virulence factor genes (e.g., vacAi1, vacAm1, cagA cagE, oipA, babA2, babB and iceA) and their association with clinical outcomes [79,80,81]. The complex association between bacterial virulence factors, alimentary variables and human immunological response determines the clinical severity of *H. pylori* infection and its complications.

*H. pylori* infection is associated with additional gastrointestinal manifestations, including haematological diseases (unexplained iron deficiency anaemia (IDA) or ITP), neurological disorders, cardiovascular diseases, skin disorders and obesity [21]. Unexplained IDA and ITP received more attention as both diseases show improvement after treatment of H. pylori infection. In Malaysia, the Malay population has the lowest prevalence of H. pylori infection compared to Indian (49.0–52.3%) and Chinese populations (26.7–57.5%) [22,82].

Gan et al. reported a lower prevalence rate of *H. pylori* infection amongst ITP patients in the country (22.0%) [22], than the overall rate reported by Goh et al. (35.9%) [82] which was comparable to North America’s reports. The prevalence rate is relatively low compared to populations in developed countries. [83,84,85,86]. Gan et al. reported a lower prevalence rate of *H. pylori* infection amongst ITP patients in the country (22.0%) [22], than the overall rate reported by Goh et al. [78] (35.9%) which was comparable to Northern America’s reports which was also low in comparison to the population in developed countries [79,80,81,82].

### 5.3. Hepatitis C Virus

Hepatitis C is a liver infection caused by the hepatitis C virus (HCV). The blood-borne virus can cause acute and chronic hepatitis, with symptoms ranging from mild to threatening diseases such as malignancy and liver cirrhosis. Hepatitis C virus infections are mostly transmitted through intravenous drug usage, transfusion of blood products and sexual activity [87,88,89,90,91]. Previous studies revealed that ITP could be secondary to HCV infection [92,93,94]. In Malaysia, the association between both diseases was only reported by Andy et al. [23]. However, the case report did not discuss ITP in HCV but instead discussed the presentation of subcutaneous mycosis in an immunocompromised patient.

## 6. Epidemiological and Treatment of Sitp Data in Malaysia: Survey Data from Hema-Tological Treatment Centers in Malaysia

As described in the previous sections, the literature search returned a limited number publications on the primary diseases that cause sITP. We have thus taken an initiative to gather the epidemiological and treatment of sITP data by forwarding sITP survey questions to four hematological centres in the country. Those questions and centres are listed in Table 5.

In general, several diseases (e.g., Kaposi’s sarcoma, dengue haemorrhagic fever (DHV), Evan’s syndrome, Wiskott-Aldrich syndrome (WAS), and infections listed in Table 1 were also the common cause of sITP in the selected hematological centres. However, drug-induced sITP, and viral hepatitis have never been reported, but were found to cause sITP in Malaysia (Table 5). Nonetheless, findings from both the literature review and survey showed that a similar treatment was given to the patients (i.e., based on either primary or secondary diseases). Therefore, a nation-wide survey should be conducted in order to obtain an accurate picture of conditions which can linked to sITP.

## 7. Conclusions

The relationship between ITP and autoimmune diseases, malignancy and infections raises questions concerning the mechanism involved in these associations. The understanding of ITP reveals that it is commonly associated with autoimmune diseases (Evan’s syndrome, SLE and WAS), malignancy (Kaposi’s sarcoma and breast cancer) and infection (DHF, *H. pylori* and HCV). The underlying mechanism for the development of ITP in secondary forms remains uncertain. The incidence or prevalence of sITP has remained unclear to the country, except for with regard to *H. pylori* infection. It has been suggested that recommendations should be made on the risk assessment and diagnosis of sITP in Malaysia, and more attention should be paid to the new diseases causing sITP, such as COVID-19. This is necessary and well supported by literature and survey data presented in this review. Future work should focus on a large scale retrospective study from multiple haematological centres to provide a better picture of epidemiology and a diagnosis of sITP in Malaysia. Presently, WAS, DHV, *H. pylori* and HCV have a specific genes/viral serology that can be used for novel strategies to diagnoses and manage diseases. However, the genetic risk factor of pITP is still underdetermined and remains unexplored. Further studies are needed to understand those exceptional but relevant associations. Moreover, there is a need to update the existing CPG of management of ITP in Malaysia. The CPG need to be revised and updated at least once every five years by including up-to-date findings on comprehensive assessments and systematic approaches of current medical practices.

## Figures and Tables

**Figure 1 healthcare-10-00038-f001:**
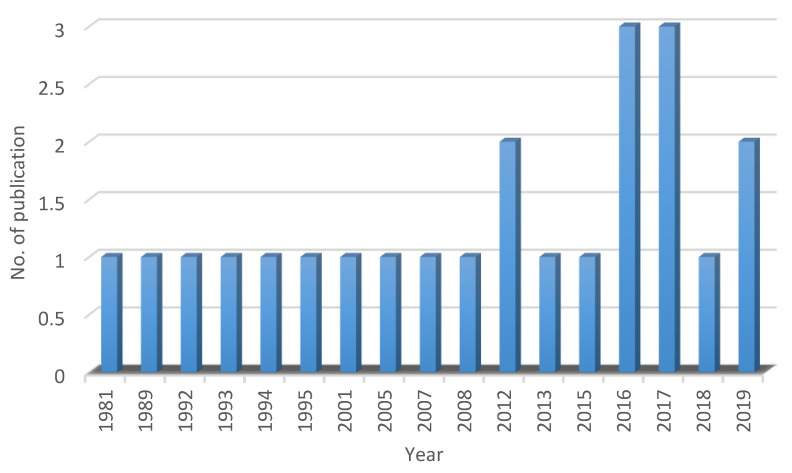
Number of sITP publications in Malaysia between 1981 and 2019.

**Table 1 healthcare-10-00038-t001:** List of sITP publications in Malaysia.

No	Author	Secondary Disease	Type
1	Pereira et al. (1981) [24]	Antiplatelet antibodies	Full study
2	Noor Akmal & Abd Wahab (1989) [16]	Kaposi’s sarcoma	Case report
3	Ng (1992) [6]	Evan’s syndrome	Original article
4	Leong & Srinivas (1993) [19]	Dengue haemorrhagic fever	Case report
5	Pasangna et al. (1994) [10]	Systemic lupus erythematosus	Case report
6	Jackson et al. (1995) [7]	Evan’s syndrome, systemic lupus erythematosus	Original article
7	Abdul Wahid et al. (2001) [17]	Breast carcinoma	Letter to editor
8	Hamidah et al. (2005) [8]	Evan’s syndrome	Case report
9	Mohd Shahrir et al. (2007) [11]	Systemic lupus erythematosus	Original article
10	Kanaheswari et al. (2008) [25]	Autosomal dominant polycystic kidney disease, congenital hepatic fibrosis	Case report
11	Palaniapan & Ramanaidu (2012) [9]	Evan’s syndrome, autoimmune hepatitis	Case report
12	Tan & Goh (2012) [21]	*Helicobacter pylori* infection	Review
13	Gan et al. (2013) [22]	*Helicobacter pylori* infection	Original article
14	Goh & Ong (2015) [12]	Systemic lupus erythematosus	Case report
15	Baharin et al. (2016) [15]	Wiskott-Aldrich syndrome	Case report
16	Lim et al. (2016) [26]	End-stage renal failure, calciphylaxis	Case report
17	Lum et al. (2016) [18]	Acute lymphoblastic leukemia	Letter to editor
18	Kuan et al. (2017) [27]	Langerhens cell histiocytosis	Case report
19	Han et al. (2017) [13]	Antiphospholipid syndrome	Case report
20	Andy et al. (2017) [23]	Hepatitis C, subcutaneous mycosis	Case report
21	Wan Jamaluddin et al. (2018) [28]	Vaccine induced, hematopoietic stem cell transplantation	Case report
22	Boo et al. (2019) [20]	Dengue haemorrhagic fever	Case report
23	Ghazali et al. (2019) [14]	Antiphospholipid syndrome, chronic vascular ulcer	Case report

**Table 2 healthcare-10-00038-t002:** Summary for ITP and Evan’s syndrome in Malaysia from 1992 to 2012.

No	Author	Year of Study	*n*	♂/♀	Condition		Type of Treatment	Other Problem
					Simultaneously	Sequentially		
1	Ng (1992) [6]	1981–1989	12	2/10	7	5	Prednisolone Danazol Splenectomy Methylprednisolone Intravenous immunoglobulin	Nehrotic syndrome Benign intracranial hypertension Pneumonia Vasculitis rash
2	Jackson et al. (1995) [7]	1984–1993	2	0/1	2		Prednisolone	Antinuclear antibodies positive Mild proteinuria
3	Hamidah et al. (2005) [8]	1998	1	0/1		1	Prednisolone Intravenous immunoglobulin Splenectomy Splenectomy	
4	Palaniapan et al. (2012) [9]	2012	1	1/0	1		Prednisolone Azathioprine	Autoimmune hemolytic hepatitis
TOTAL			16	3/12	10	6		

♂—male, ♀—female.

**Table 3 healthcare-10-00038-t003:** Status of treatment ITP patients with Evan’s syndrome in Malaysia.

No	Author	*n*	Treatment Response (%)					Died (%)
			Default	Prednisolone	Danazol	Splenectomy	Methylprednisolone	
1	Ng (1992) [6]	12	1	2	1	3	1	4
2	Jackson et al. (1995) [7]	2		1				
3	Hamidah et al. (2005) [8]	1				1		
4	Palaniapan et al. (2012) [9]	1		1				
TOTAL		16	1 (6.25)	4 (25.00)	1 (6.25)	4 (25.00)	1 (6.25)	4 (25.00)

**Table 4 healthcare-10-00038-t004:** Summary for ITP and SLE in Malaysia from 1994 to 2015.

No	Author	*n*	♂/♀	Age	Treatment	Other Problem
1	Pasangna et al. (1994) [10]	1	1/0	8	Bacampicillin Amoxycilinl Prednisolone	Evan’s syndrome
2	Jackson et al. (1995) [7]	1	0/1	45	Prednisolone Cyclophosphamide	Renal and cerebral involvement
3	Mohd Shahrir et al. (2007) [11]	2	1/1	14, 26	Rituximab Steroids Mycophenolate mofetil Cyclosporin A	
4	Goh & Ong (2015) [12]	1	0/1	22	Intravenous methylprednisolone Intravenous immunoglobulin Azathioprine Intravenous cyclophosphamide	Systemic sclerosis Subdural hematoma
TOTAL		5	2/3			

♂—male, ♀—female.

**Table 5 healthcare-10-00038-t005:** sITP survey questions to hematological centres in Malaysia.

No.	Questions	Hospital Universiti Sains Malaysia (A.D. Abdullah, Personal Communication)	Hospital Ampang, Kuala Lumpur (Wong T.G. Personal Communication)	Hospital Raja Perempuan Zainab II, Kota Bharu (Nuruaini S.A.S. Personal Communication)	Hosp Sultanah Aminah, Johor Bahru (Lim S.M. Personal Communication)
1.	What are the causes of sITP in your centres?	Sepsis related, SLE, drug induced thrombocytopenia (i.e., antibiotics, heparin) and post-transfusion on purpura	SLE, drug induced, infection, lymphoproliferative disease, viral hepatitis and vaccine-induced	SLE, lymphoproliferative disorder, chronic liver disease with portal hypertension and drug induced	Drug induced, lymphoprolif-erative disease, SLE, infection, and vaccine-induced
2.	How is the diagnosis of ITP is made?	Adult patient presenting with severe thrombocytopenia is actively screened for systemic lupus erythematosus (SLE) or adult presentation of congenital thrombocytopenia. Other secondary thrombocytopenias may have other easily identifiable accompanying features	Diagnosis by exclusion	Will do appropriate blood test including bone marrow aspiration and imaging to confirm secondary causes of ITP	Investigate the primary causes
3.	Is it the primary diseases that cause sITP determine diagnosis and treatment?	Yes	yes	yes	Yes
4.	What are treatment cycle of sITP and treatment difficulty?	Patient with suspected ITP will be treated with pulse dexamethasone, whereas those suspected to have underlying SLE will receive prednisolone as the initial treatment. These patients will be kept on hydroxychloroquine and low dose prednisolone for at least two years. Secondary thrombocytopenias may have other easily identifiable accompanying features and is treated accordingly, defined by the primary illness	Treatment of the primary cause will resolve the thrombocytopenia. Refractory ITP may be difficult to treat	Treatment of secondary ITP is not difficult as most of the time, the thrombocytopenia will resolve after treatment of the primary causes	Secondary ITP may not respond well to the usual immunosuppression and need to identify the underlying disorder. Treating the primary disease and the thrombocytopenia may improve

## Data Availability

The data presented in this study are available on request from the corresponding author.

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
