# Peer review of "A Review on Secondary Immune Thrombocytopenia in Malaysia"

_healthcare, 2021, doi:10.3390/healthcare10010038_

Round 1
Reviewer 1 Report
1. Table 2, 3 and 4, Is the data assembled is from Malaysia. In title it shas been specified as Secondary Immune Thrombocytopenic Purpura in Malaysia.
2. Table 2 and 3, data is till 2012. Table 4, data is till 2015.
3. Table 1. The outcome of the studies should aslo be included. The data is till 2019.
4. What is the significance of figure 1. Outcome should be highlighted.
5. Conclusive summary and future direction should also be included.
Author Response
Reviewer 1
|
Comment |
Action |
1 |
Table 2, 3 and 4, Is the data assembled is from Malaysia. In title it has been specified as Secondary Immune Thrombocytopenic Purpura in Malaysia. |
Thank you for your question. Yes, Tables 2-4 are ITP publications in Malaysia. In those tables, we tabulated the summary of reported ITP cases along with their secondary causes; i.e., Tables 2-3 for Evan’s syndrome, Table 4 for SLE.
|
2 |
Table 2 and 3, data is till 2012. Table 4, data is till 2015. |
Yes, the data was tabulated as in the text. We want to clarify that, a total of 23 ITP publications were published in Malaysia between 1981 and 2019 [Page 2, Paragraph 3]. Of the numbers, we summarised the publications into condition groups that cause ITP. We then decided to discuss top three groups of condition (have more than 1 publication) which are autoimmune disease, malignancy and infection.
In autoimmune disease, three diseases were discussed and such data was extracted from Table 1. · Evan’s syndrome (Tables 2-3) – 4 publications that published between 1992 and 2012, · SLE (Table 4) – 4 publications that published between 1994 and 2015.
|
3 |
Table 1. The outcome of the studies should also be included. The data is till 2019. |
Thank you for your suggestion. We tabulated 23 secondary ITP articles in Table 1 from where 15 diseases were reported associated with ITP [Page 2, Paragraph 4].
Yes, the latest paper was published in 2019.
|
4 |
What is the significance of figure 1. Outcome should be highlighted. |
Figure 1 summarise the frequency of sITP publications and outcomes discussed in Page 2, Paragraph 3 [Lines 61-74]. |
5 |
Conclusive summary and future direction should also be included. |
Done |

Reviewer 2 Report
The authors have prepared a review of sITP literatures published in Malaysia for past four decades. The English of the paper is easily understandable. However, the content of the innovation is insufficient.
- This review provides examples of several primary diseases that causing sITP in Malaysia. However, there have been many reviews on the primary diseases that cause sITP, and the authors' selection of several diseases is not comprehensive enough. Is there incomplete data for the lack number of publications of other primary diseases mentioned by the author? For example, sITP is not a rare clinical disease and some of the cases may not been reported. Would it be more convincing for the authors to collect data from large hematological treatment centers than to simply search the reported literature?
- As for the description of sITP, most of the content is about the primary disease itself and the treatment after the occurrence of sITP, but the discussion of sITP is not in-depth enough. It is suggested that the diagnosis, treatment, treatment difficulty and treatment cycle of sITP caused by different primary diseases should be discussed in terms of more clinical treatment suggestions for the readers.
- The discussion is not enough in the conclusion. It is suggested that recommendations should be made on the risk assessment and diagnosis of sITP in Malaysia, and more attention should be paid to the new diseases causing sITP, such as COVID-19.
Author Response
Reviewer 2
|
Comment |
Action |
1 |
The authors have prepared a review of sITP literatures published in Malaysia for past four decades. The English of the paper is easily understandable. However, the content of the innovation is insufficient.
This review provides examples of several primary diseases that causing sITP in Malaysia. However, there have been many reviews on the primary diseases that cause sITP, and the authors' selection of several diseases is not comprehensive enough. Is there incomplete data for the lack number of publications of other primary diseases mentioned by the author? For example, sITP is not a rare clinical disease and some of the cases may not been reported. Would it be more convincing for the authors to collect data from large hematological treatment centers than to simply search the reported literature?
|
We appreciate your valuable comments and has now include coverage of these points in the revised version of our manuscript. Agreed, many sITP cases were not reported and for that reason we have taken an initiative to get information from 4 hematological treatment centers in the country – refer section 6 and Table 5.
|
2 |
As for the description of sITP, most of the content is about the primary disease itself and the treatment after the occurrence of sITP, but the discussion of sITP is not in-depth enough. It is suggested that the diagnosis, treatment, treatment difficulty and treatment cycle of sITP caused by different primary diseases should be discussed in terms of more clinical treatment suggestions for the readers.
|
We thank the reviewer for making these valuable suggestions. Suggestions from the Reviewer are now included in the revised version of our manuscript and please refer newly added section (#6) and Table 5 |
3 |
The discussion is not enough in the conclusion. It is suggested that recommendations should be made on the risk assessment and diagnosis of sITP in Malaysia, and more attention should be paid to the new diseases causing sITP, such as COVID-19.
|
Done and please refer Lines 312-314. |

Reviewer 3 Report
The authors have conducted a "review on secondary immune thrombocytopenic purpura in Malaysia". To my view, the paper is not a significant contribution to the field of knowledge and does not warrant publication. I was not able to identify the novelty of the paper because the paper is poorly constructed. Of course it would be interesting to gather epidemiological data on the status of secondary ITP in Malaysia but definitely this cannot be done using isolated case reports and just a few conditions which can be linked with sITP. Firstly, the paper would require a major polishing of the English and scientific language used. This is a not a newspaper article and we should avoid writing in layman's terms, e.g., "problem" instead of "disease", disorder etc.Secondly, you only reviewed a database (MEDLINE) - of note, PubMed is the search service and not the database itself. How about other databases? Web of Science, SCOPUS, ScienceDirect etc and especially Chinese databases should have been screened. Thirdly, you state this is a study on sITP and Malaysia but discuss in general about the subject. I don't see the novelty of the paper, definitely hematology experts have reviewed the pathogenesis and management of sITP. Moreover, even if you did, the data presented in the paper is of local importance. You didn't review other causes of sITP: CLL, lymphoma, vaccines, drug-induced sITP, pediatric ITP, HBV etc. The references are not formatted according to the style required by the journal - American Chemical Society. In addition, I don't see the link between the paper and the special issue it was submitted to.
Moreover, the correct denomination is immune thrombocytopenia and not immune thrombocytopenic purpura - this term has been dropped by experts.
Author Response
Reviewer 3
|
Comment |
Action |
1 |
The authors have conducted a "review on secondary immune thrombocytopenic purpura in Malaysia". To my view, the paper is not a significant contribution to the field of knowledge and does not warrant publication. I was not able to identify the novelty of the paper because the paper is poorly constructed. Of course it would be interesting to gather epidemiological data on the status of secondary ITP in Malaysia but definitely this cannot be done using isolated case reports and just a few conditions which can be linked with sITP.
|
Noted with thanks and please refer to our responses to comment #1 and #2 given by the Reviewer #2. |
2 |
Firstly, the paper would require a major polishing of the English and scientific language used. This is a not a newspaper article and we should avoid writing in layman's terms, e.g., "problem" instead of "disease", disorder etc.
|
Thank you and we have to our best rectified this issue. We are willing to send out this manuscript for English editing if needed.
|
3 |
Secondly, you only reviewed a database (MEDLINE) - of note, PubMed is the search service and not the database itself. How about other databases? Web of Science, SCOPUS, ScienceDirect etc and especially Chinese databases should have been screened. |
Thank you for highlighting this point. For your information, data mining was conducted using several search engines (i.e., Web of Science, SCOPUS and ScienceDirect). All sITP in Malaysia was found in the Medline. That why we only mentioned about Medline in the original version of our manuscript. However, we have revised the section and please refer Lines 62-63. |
|
Thirdly, you state this is a study on sITP and Malaysia but discuss in general about the subject. I don't see the novelty of the paper, definitely hematology experts have reviewed the pathogenesis and management of sITP. Moreover, even if you did, the data presented in the paper is of local importance. You didn't review other causes of sITP: CLL, lymphoma, vaccines, drug-induced sITP, pediatric ITP, HBV etc.
|
Agreed, this review focuses on ITP in Malaysia. As explained earlier, we have now included survey data from 4 hematological treatment centers in Malaysia. These hematological treatment centers located in the sourt, east and west parts of the Peninsular Malaysia and are thus may capture other causes (e.g., CLL, lymphoma, vaccines, drug-induced sITP etc.) and treatments of sITP.
We believe that this review may also benefit ITP related managements in countries which share a common epidemiological profiles, at least for medical treatment.
|
|
The references are not formatted according to the style required by the journal - American Chemical Society. In addition, I don't see the link between the paper and the special issue it was submitted to.
|
We have formatted List of References according to the style adopted by the Healthcare journal.
One of the very common symptoms of ITP is skin disorder and hope that this review suite well to this special issue in Healthcare. |
|
Moreover, the correct denomination is immune thrombocytopenia and not immune thrombocytopenic purpura - this term has been dropped by experts.
|
We agree with the Reviewer and the correct terminology is adopted in the revised version of our manuscript [Page 1, Paragraph 2]. |

Round 2
Reviewer 1 Report
The paper is improved and well presented now.
Author Response
Reply to the reviewer’s comments: Reviewer #1
Comment Number |
Reviewer’s comment |
Reply by the author(s) & Changes done on page number |
1.
|
The paper is improved and well presented now. |
Thank you. We were gratified to receive constructive comments from the Reviewers who have helped to improve the earlier version of our manuscript. |

Reviewer 2 Report
The author has revised the previous comments, but the last one is too simple.
Author Response
Reply to the reviewer’s comments: Reviewer #2
Comment Number |
Reviewer’s comment |
Reply by the author(s) & Changes done on page number |
1 |
The author has revised the previous comments, but the last one is too simple. |
Thank you for your further advice and please refer p. 11, lines 311-316.
|

Reviewer 3 Report
The authors did not make an effort to answer to my comments. The revision of the paper was poorly handled and I still do not believe it is suitable for publication. I appreciate they added a new section but the construct and novelty of the paper remain poor. I suggested changing the term immune thrombocytopenic purpura to immune thrombocytopenia but the authors did not bother to change the title as well. I understand that secondary ITP might be an interesting topic in Malaysia but the design of the paper should either have been that of a multicentric observational retrospective study from multiple hematology centers in your country +/- survey research + review of the literature.
Author Response
Reply to the reviewer’s comments: Reviewer #3
Comment Number |
Reviewer’s comment |
Reply by the author(s) & Changes done on page number |
1. |
The authors did not make an effort to answer to my comments. The revision of the paper was poorly handled and I still do not believe it is suitable for publication. I appreciate they added a new section but the construct and novelty of the paper remain poor. I suggested changing the term immune thrombocytopenic purpura to immune thrombocytopenia but the authors did not bother to change the title as well. I understand that secondary ITP might be an interesting topic in Malaysia but the design of the paper should either have been that of a multicentric observational retrospective study from multiple hematology centers in your country +/- survey research + review of the literature. |
Thank you for your further comments on our manuscript. We have revised the manuscript, including on title page. As suggested, we have included literature and survey data on ITP in this review. However, multicentric observational retrospective study from multiple hematology centers seem to be unrealistic and technically cannot be done during this revision and not suitable for the submitted review paper. However, we has now include coverage of these points in the revised version of our manuscript – p. 11, lines 315-316. |
